# *Pasteurella* sp. associated with fatal septicaemia in six African elephants

Chris M. Foggin[1,14], Laura E. Rosen [2,13,14] ✉, Marijke M. Henton[3], Angela Buys[4], Toby Floyd[5], Andrew D. Turner [6], Jonathan Tarbin[7], Antony S. Lloyd [7], Columbas Chaitezvi[8], Richard J. Ellis[9], Helen C. Roberts [10], Akbar Dastjerdi [11], Alejandro Nunez [5], Arnoud H. M. van Vliet [12] & Falko Steinbach [11,12]

The sudden mortality of African elephants (*Loxodonta africana*) in Botswana and Zimbabwe in 2020 provoked considerable public interest and speculation. Poaching and malicious poisoning were excluded early on in the investigation. Other potential causes included environmental intoxication, infectious diseases, and increased habitat stress due to ongoing drought. Here we show evidence of the mortalities in Zimbabwe as fatal septicaemia associated with Bisgaard taxon 45, an unnamed close relative of *Pasteurella multocida*. We analyse elephant carcasses and environmental samples, and fail to find evidence of cyanobacterial or other intoxication. Post-mortem and histological findings suggest a bacterial septicaemia similar to haemorrhagic septicaemia caused by *P. multocida*. Biochemical tests and 16S rDNA analysis of six samples and genomic analysis of one sample confirm the presence of Bisgaard taxon 45. The genome sequence contains many of the canonical *P. multocida* virulence factors associated with a range of human and animal diseases, including the pmHAS gene for hyaluronidase associated with bovine haemorrhagic septicaemia. Our results demonstrate that Bisgaard taxon 45 is associated with a generalised, lethal infection and that African elephants are susceptible to opportunistically pathogenic *Pasteurella* species. This represents an important conservation concern for elephants in the largest remaining metapopulation of this endangered species.

African elephants (*Loxodonta africana*) are a flagship species under significant pressure from poaching and habitat loss[1]. They were listed as an endangered species on the IUCN Red List in 2021[2], and elephant populations have decreased by 144,000 from 2007–2014 to ~350,000, with ongoing losses estimated at 8% annually[1]. Protected areas of elephant habitat and wildlife corridors between them are essential to the long-term survival of the species[3]. Conservation of wildlife habitat and biodiversity are among the goals of Transfrontier Conservation

[1]Victoria Falls Wildlife Trust, Victoria Falls, Zimbabwe. [2]Transboundary Epidemiology Analytics, LLC, Alpharetta, GA, USA. [3]Vetdiagnostix, Blue Hills, Midrand, South Africa. [4]Design Biologix, Erasmusrand, Pretoria, South Africa. [5]Pathology and Animal Sciences Department, Animal and Plant Health Agency Weybridge, Addlestone, Surrey KT15 3NB, UK. [6]Centre for Environment Fisheries and Aquaculture Science, The Nothe, Weymouth, Dorset DT4 8UB, UK. [7]Fera Science, Biotech Campus, York YO41 1LZ, UK. [8]Zimbabwe Parks & Wildlife Management Authority, Harare, Zimbabwe. [9]Surveillance and Laboratory Services Department, Animal and Plant Health Agency Weybridge, Addlestone, Surrey KT15 3NB, UK. [10]Department for Environment Food & Rural Affairs, Nobel House, 17 Smith Square, London SW1P 3JR, UK. [11]Virology Department, Animal and Plant Health Agency Weybridge, Addlestone, Surrey KT15 3NB, UK. [12]Department of Comparative Biomedical Sciences, School of Veterinary Medicine, Faculty of Health and Medical Sciences, University of Surrey, Guildford GU2 7AL, UK. [13]Present address: Victoria Falls Wildlife Trust, Victoria Falls, Zimbabwe. [14]These authors contributed equally: Chris M. Foggin, Laura E. Rosen. ✉ e-mail: kazaepi@gmail.com

Areas (TFCAs), which are conservation initiatives which encompass private and communal land as well as parks and game reserves across country borders[4].

The Kavango–Zambezi (KAZA) TFCA, encompassing over 500,000 km$^2$ of land across Botswana, Zimbabwe, Zambia, Angola, and Namibia, is the largest Transfrontier Conservation Area in the world and is home to the biggest contiguous population of elephants in Africa[5]. The KAZA Elephant Survey in 2022 estimated a population of 227,900 elephants in the TFCA, with nearly 90% concentrated in Botswana and Zimbabwe (58% and 29%, respectively)[5]. Based on surveys conducted in 2014–2015, these two countries have the highest densities of elephants compared to all other African countries[1]. The metapopulation of elephants in KAZA is therefore crucial to the conservation of the species.

Episodic mortality of elephants has occurred in north-western Zimbabwe during past dry seasons (April–October), often as a result of drought[6] or anthrax[7]. Malicious poisoning using cyanide, illegally sourced from the gold mining industry in Zimbabwe, has also caused elephant mortality in recent years[8]. In the hot dry season (September–October) of 2019, at least 200 elephants in the area died as a result of drought and starvation[9]. Occasional scattered elephant mortalities have occurred since January 2020, and over the course of late August to November 2020, a mortality event of 35 elephants occurred in north-western Zimbabwe. This most recent event was of particular interest given that it was preceded by a mass mortality event involving ~350 elephants in neighbouring northern Botswana from May to June 2020[10,11]. Common causes of elephant mortality, including poaching and anthrax, were ruled out in Botswana[12], and the deaths have been attributed to an unspecified cyanobacterial toxin[13].

Here, we show findings that elephant mortalities in Zimbabwe in 2020 were due to bacterial septicaemia associated with infection with *Pasteurellaceae* Bisgaard taxon 45, an unnamed close relative of *Pasteurella multocida*.

## Results

### Epidemiology of elephant deaths

A total of 35 African elephants were found dead in north-western Zimbabwe: 34 elephants were found from 24 August–20 September 2020, and a single elephant was found on 9 November 2020 (Fig. 1). The total area over which dead elephants were identified was ~40 × 25 km. The area encompasses the Woodlands Safari and Resettlement Area, Matetsi Unit 7 Safari Area, Pandamasuie State Forest, and the Zambezi National Park. At least 11 elephants died within ~24 h in an area of ~50 km$^2$ in the Pandamasuie State Forest. Two elephants were found within 100 m of a water point. The estimated age of dead elephants ranged from 18 months–30 years (median = 10 years). Elephants of both sexes were found dead (16 males, 9 females). Age and sex were not determined in all cases, as some carcasses had already decomposed and/or been scavenged, and others were identified aerially in remote areas where timely ground searches could not be performed. No dead scavengers or other wildlife species were reported or observed in the vicinity of dead elephants as would be expected with cyanide or other intentional poisoning. No elephants had their tusks removed from poaching, and no external signs of trauma were observed.

### Evidence for bacterial septicaemia

The carcasses of five elephants were further investigated using postmortem, gross pathology and histopathology techniques, as well as toxicology analysis, with full details being provided in the Supplementary Information (with summaries of gross pathology and histopathology in Supplementary Table 1 and summaries of laboratory testing in Supplementary Table 2). Briefly, the carcasses were in average body condition with hepatomegaly and splenomegaly as the most prominent gross pathological findings, with variable haemorrhages

(Fig. 2) across the epicardium, liver, lungs, intestinal serosae, hepatic and splenic lymph nodes, and in one case, the diaphragm. Histopathological lesions in elephants were similar and consisted of acute multifocal heterophilic and necrotizing inflammation in liver, spleen, and lymph node, with presence of intralesional Gram-negative bacterial colonies of coccobacillary morphology (Fig. 3). Specifically, elephant VF20/112 displayed necrotizing lesions in spleen and liver, with the additional presence of fibrinocellular and bacterial emboli in the pulmonary vasculature. Presence of Gram-negative bacterial colonies without associated morphological changes was observed in veins and capillaries, prominently in the encephalon. Acute multifocal heterophilic and necrotizing lymphadenitis, hepatitis and splenitis with intralesional Gram-negative coccobacilli was observed in elephant VF20/113. Most blood smears ($n = 13/15$) stained with Giemsa contained small to moderate numbers of bacteria with a bipolar, short-rod, or coccobacilli morphology (0.5–2 μm), and intracellular bacteria were observed (Table 1).

Toxicology analyses were performed to investigate the role of poisoning, as suggested elsewhere[13]. There was an absence of cyanide, saxitoxin analogues, tetrodotoxin analogues, anatoxin-a or cylindrospermopsin. While non-targeted toxicological analysis provided a range of incidental findings, no result was significant or considered relating to the clinical picture. Mass spectrometry of stomach contents did not show the presence of toxins or associated breakdown products. In summary, the pathological findings were consistent with bacterial septicaemia and bacteraemia at the time of death (Supplementary Notes).

### Molecular evidence of Bisgaard taxon 45

All the samples [brain, liver and spleen] from elephant VF20/112 yielded a heavy growth of a putative *Pasteurella* sp., while *B. anthracis* or other possible pathogens were not detected. Two isolates obtained from the brain and liver had their 16S rDNA gene sequence compared with other listed sucrose negative isolates[14], which confirmed them to be a *Pasteurella* sp., with the 16S rDNA sequence most closely resembling that of Bisgaard taxon 45 (Fig. 4). The biochemical tests showed that the isolates also corresponded phenotypically with taxon 45 of Bisgaard, which has been reported to be closely related to unusual, sucrose negative *P. multocida* strains[14]. Illumina sequencing from elephant VF20/112 samples followed by removal of *L. africana* sequences allowed for genome reconstruction of the VF20HR isolate, which had a 2.2 Mbp genome, and comparison of the *infB* and *rpoB* gene fragments confirmed its assignment to the Bisgaard taxon 45 (Fig. 4). Further comparative genomics analyses showed that the VF20HR genome clustered just outside *P. multocida* genomes, but did not cluster with other *Pasteurella* spp. Analysis for the presence or absence of *Pasteurella* virulence factors (Supplementary Fig. 1) showed that the VF20HR genome encoded contained 22 of the 25 virulence factors tested for[15], including the pmHAS hyaluronidase which is linked to *P. multocida* type B isolates causing bovine haemorrhagic septicaemia[16]. Other *P. multocida* virulence factors such as adhesion factors and iron acquisition systems were also detected, confirming the close association of VF20HR to *P. multocida*. The bacteriological and genetic findings were supported by the detection of Bisgaard taxon 45 in 10 FTA cards from 6/14 elephants tested via 16S rDNA PCR analysis (Table 1). This included elephant VF20/112, which was previously positive on bacterial culture and 16S rDNA analysis. No viruses were detected by either microarray or WGS analysis using samples from VF20/112 and VF20/113 (Supplementary Methods).

## Discussion

African elephants are declining in numbers, and every significant mortality event gives rise to concern and speculation[11]. Elephant mortalities across northern Zimbabwe and neighbouring northern Botswana in 2020 were widely reported, and while the mortalities in Botswana have

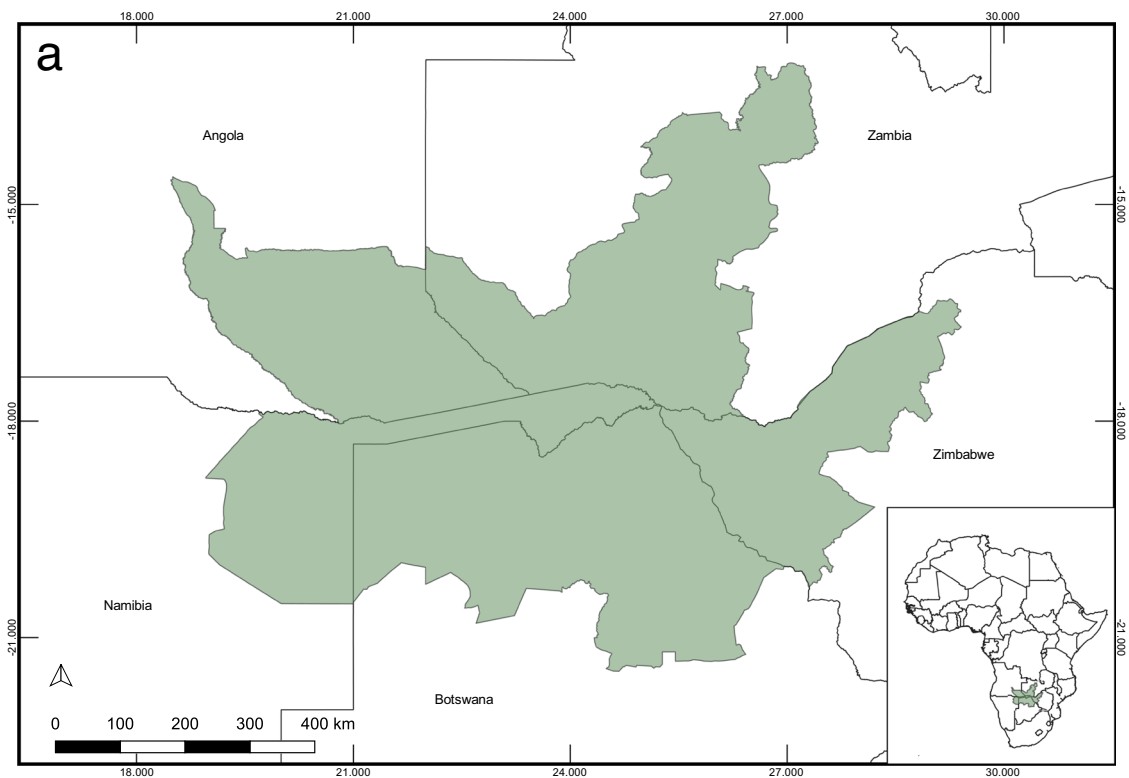

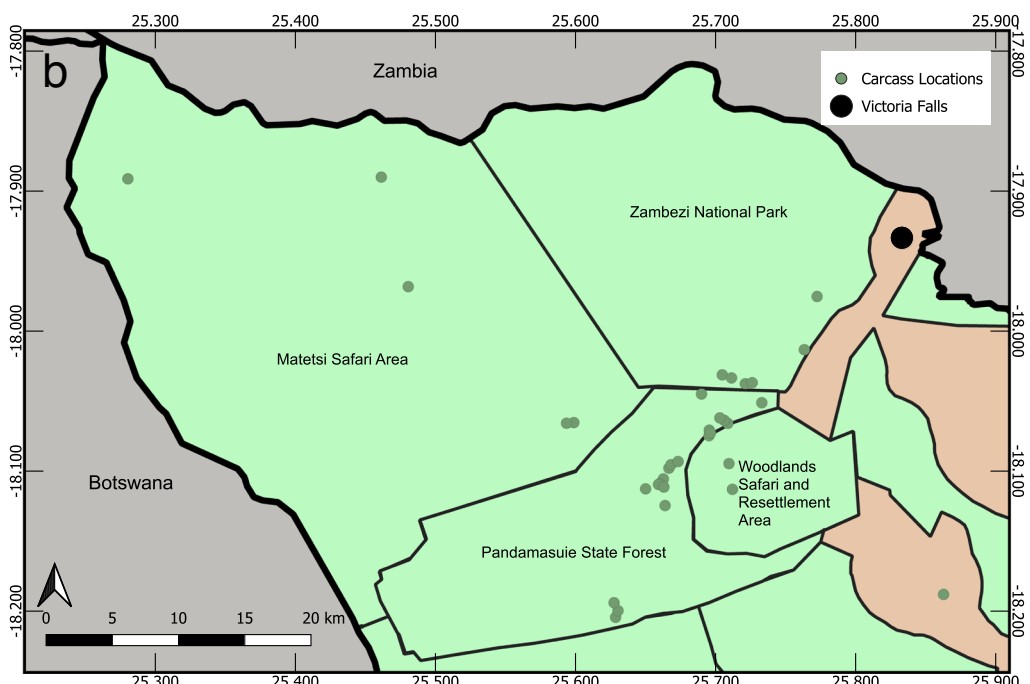

**Fig. 1 | Maps of the Kavango–Zambezi Transfrontier Conservation Area (KAZA TFCA) and elephant carcass locations. a** The Kavango–Zambezi Transfrontier Conservation Area (KAZA TFCA), shown in green, spans portions of Botswana, Namibia, Angola, Zambia, and Zimbabwe. The KAZA TFCA shapefile was used with permission from Peace Parks Foundation. **b** A more detailed map of the area in north-western Matabeleland, Zimbabwe where elephant carcasses were found, including the approximate location of the carcasses. Protected areas in Zimbabwe are shown in green.

been attributed to cyanobacterial neurotoxins[13], further details have not been published. This report describes findings from the elephant mortalities in Zimbabwe, which suggest that Bisgaard taxon 45, an unnamed close relative of *Pasteurella multocida*, was associated with the die-off. Of 15 sampled elephants, six showed molecular evidence of septicaemic infection by Bisgaard taxon 45, and this was corroborated by bacterial isolation and in-depth genetic analysis. There was no evidence of toxins, including those from cyanobacteria, or for any viral

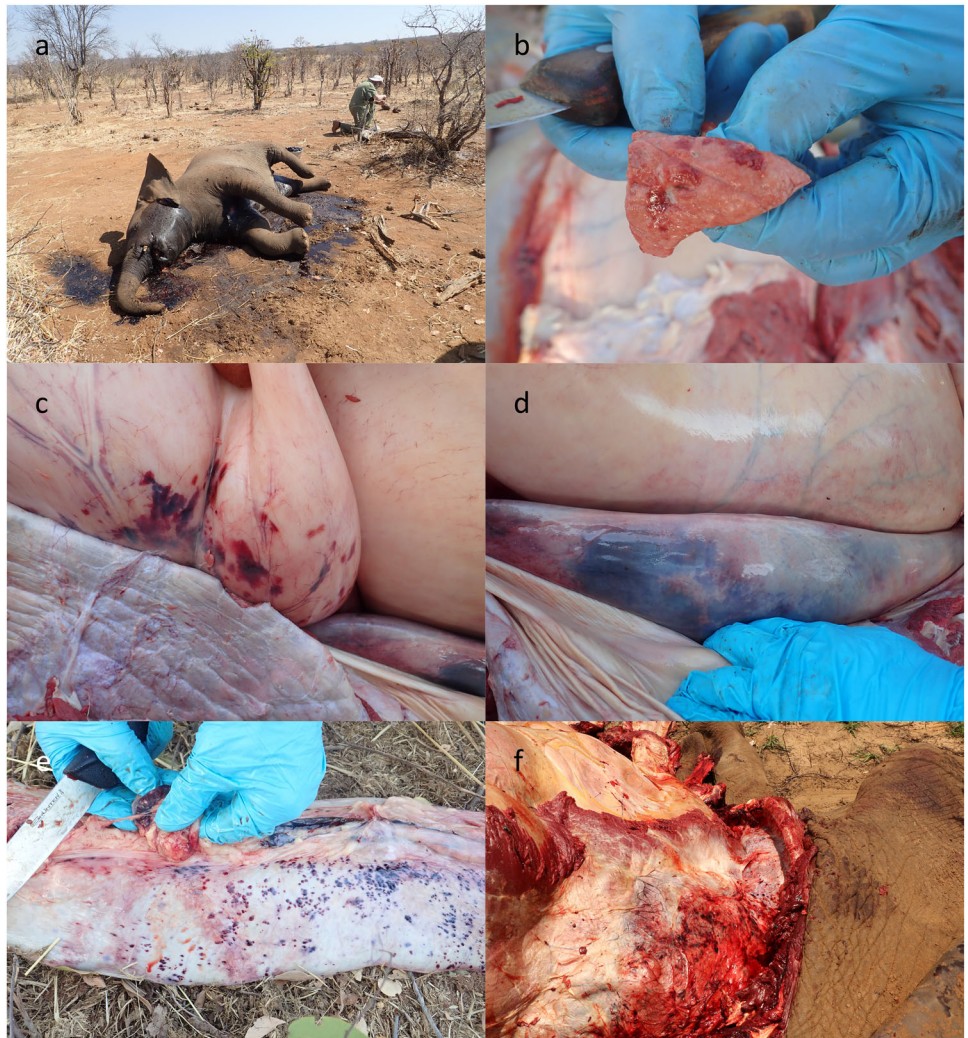

**Fig. 2 | Gross pathology findings of the dead African elephants. a** Elephant VF120A carcass in situ, showing postmortem condition with similar presentation to anthrax; only blood smears were collected (**b**) Elephant VF20/112 lung tissue showing small areas of consolidation and hyperaemia. **c** Elephant VF20/113 stomach with haemorrhages on gastric serosa; lung in foreground and liver on bottom right shown in next panel. **d** Elephant VF20/113 liver in situ adjacent to stomach, with rounded border indicating gross swelling, and parenchymal hyperaemia. **e** Elephant VF20/113 spleen, removed, with multiple capsular haemorrhages and an enlarged splenic lymph node. **f** Elephant VF20/170 thoracic diaphragm and lung with haemorrhages.

infection. Our findings add bacterial septicaemia mortalities in elephants to the known associations with Bisgaard taxon 45, which has previously been isolated from tiger and lion bite wounds in humans, a chipmunk[14], and healthy captive psittacines[17].

Here, Bisgaard taxon 45 was isolated in heavy growth from the brain, liver and spleen of elephant VF20/112, demonstrating a septicaemic condition. Older phenotypic tests did not adequately differentiate between the large variety of *Pasteurellaceae* spp.[18], and Magne Bisgaard lent his name to various groups of bacteria of the *Pasteurellaceae*, naming them as Bisgaard taxon followed by a number. Further descriptions and naming of many of these new species was only possible after the development of molecular techniques such as 16S sequencing and PCR typing[15]. However, Bisgaard et al raised concerns about the sole use of genomic DNA analysis to name bacterial species and the validity of naming species based on a single isolate[19]. Our genome sequence of a member of the Bisgaard taxon 45 is closely related to *P. multocida* and shares many, but not all, of its virulence factors.

Although bacterial septicaemia cases have not previously been associated with Bisgaard taxon 45, they may represent an ongoing phenomenon in this region. For example, samples were submitted from 14 dead elephants, several of which died suddenly, found in Hwange National Park between August–October 2019, (Supplementary Table 3). No bacterial culture was performed at the time, but after recently reviewing slides from these animals, the lesions in liver and/or spleen of three elephants are suggestive of bacterial septicaemia and resemble those observed in the 2020 mortalities. One month prior to the onset of the 2020 mortality event and -150 km southeast, a 3 month-old elephant calf was found dead and tissues were submitted for histopathology. Pathologic changes in the brain, liver, and spleen, including the presence of bacterial colonies, were similar to those reported in this study. The massive bacterial overgrowth observed in the two elephant brains examined was particularly striking, and it may be helpful to collect brain samples for histopathology where Bisgaard taxon 45 is suspected as a cause of death. Past cases may have been missed because for mortalities that were suspected to be anthrax at the time of post-mortem examination, samples for histopathology or bacterial culture were not collected and therefore not available to test if the elephants were found to be negative for anthrax. However, anthrax cannot be ruled out in the field, and the value of such samples must be weighed against the risk of opening a carcass that may release anthrax spores.

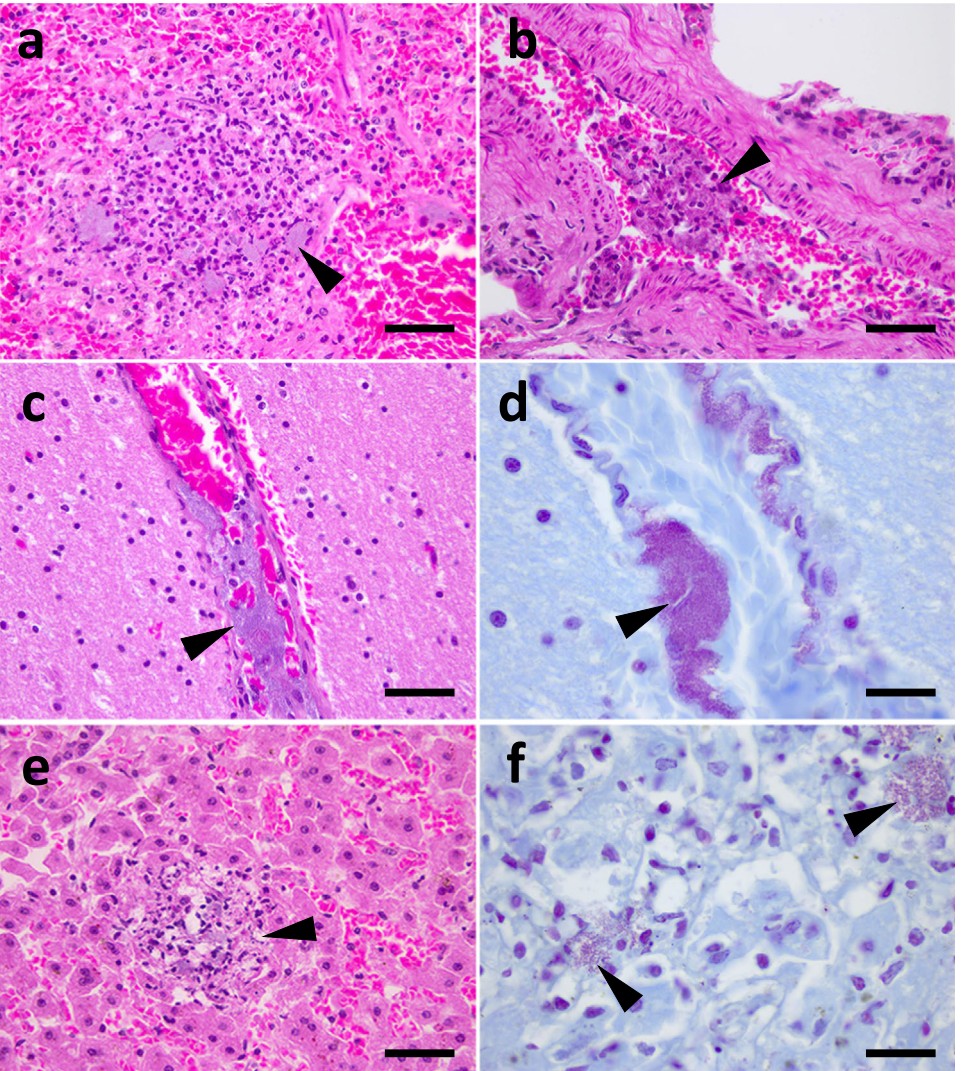

**Fig. 3 | Histopathology of lesions in tissues from two elephants. a–d** Are photomicrographs of lesions in tissues from elephant VF20/112 demonstrating foci of necrosuppurative inflammation in the spleen (**a**) with intralesional bacterial colonies (arrowhead), and fibrinocellular and bacterial emboli in the lung (**b**) and brain (**c**, **d**). Panels **e** and **f** are photomicrographs of lesions in tissues from elephant VF20/113, demonstrating foci of necrosuppurative inflammation in the liver (**e**) containing gram-negative coccobacilli (arrowheads, **e**, **f**). **a**, **b**, **c**, and **e** Haematoxylin and Eosin, x40 magnification, bar 50 microns. **d**, **f** Gram, x100 magnification, bar 20 microns. Data are representative of lesions observed from these two elephants at three independent laboratories.

The failure to identify Bisgaard taxon 45 in samples from all 15 elephants is likely due sample quality and delays in testing, but unfortunately it was not possible to obtain permits for additional culture samples in a timely manner. Most carcasses were degraded at the time of sampling, making the initial sample quality poor. Additionally, exporting wildlife samples for analysis involves obtaining multiple permits from different entities—a process which can take months[20]. Such delays preclude the use of time-sensitive assays such as bacterial culture and postpone follow-up diagnostic testing indicated by initial results. As an example, samples from elephant VF20/129 were first sent to an in-country laboratory for culture, which did not require additional time to obtain import and export permits. The *P. multocida* result from this laboratory was likely Bisgaard taxon 45, which would have been indistinguishable from *P. multocida* when serotyping or biochemical tests were not attempted. It took 32 days to acquire permits to send samples to South Africa, and samples were only received at the laboratory 8 weeks after collection, which is the likely reason for the failure to isolate Bisgaard taxon 45 there. Only *Klebsiella pneumoniae* was isolated from VF20/129 at this laboratory, where a single colony was isolated from the lung sample and on one of three initial

media on which it was inoculated. This suggests that there were few *Klebsiella* bacteria present. The *Klebsiella* finding was likely incidental as *Klebsiella* are part of the normal flora. The 16S RNA results support the presence of a mixed culture, presumed to be Bisgaard taxon 45 and *Klebsiella pneumoniae*.

After anthrax had been ruled out, haemorrhagic septicaemia was suspected based on the clinical presentation, the results of blood smears, and the isolation of bacteria first assumed to be *P. multocida*. Haemorrhagic septicaemia has previously been reported in various species, including Asian elephants (*Elephas maximus*) in Sri Lanka, India, Thailand, and Myanmar[21–25], but not African elephants. Wildlife mass mortality events have been associated with *P. multocida*-induced haemorrhagic septicaemia, most notably the 2015 die-off involving ~200,000 critically endangered saiga antelope (*Saiga tatarica tatarica* and *S. tatarica mongolica*)[26].

The epidemiology of Bisgaard taxon 45 is currently unknown, and septicaemia such as that seen with *P. multocida* has not been previously reported with this organism, although they share many of the same virulence factors. Outbreaks of haemorrhagic septicaemia in cattle are generally associated with wet, humid weather[27], while the mass

**Table 1 | Post-mortem sample results from elephants found dead in north-western Zimbabwe, 2020**

| Animal ID | Date Sampled | Sex/Age (yr) | Blood smear | Culture | 16 S rDNA | GenBank Accession No.[a] |
|---|---|---|---|---|---|---|
| VF20/112 | 24/8/2020 | M/8 | 4+ paired CB | *P. multocida*[b] Bisgaard taxon 45[c] | Bisgaard taxon 45 | AY683487.1 |
| VF20/113 | 25/8/2020 | M/8 | 4+ ALO | ND | Bisgaard taxon 45 | AY683487.1 |
| VF20/114 | 25/8/2020 | M/25 | 3+ paired CB*; few ALO | ND | NA | |
| VF20/115 | 25/8/2020 | M/UNK | 2+ paired CB; PBR | ND | Bisgaard taxon 45 | AY683487.1 |
| VF20/116 | 25/8/2020 | M/6 | 4+ paired CB; few ALO | ND | Bisgaard taxon 45 | AY683487.1 |
| VF20/120A | 26/8/2020 | F/6 | 2+ CB; 5+ PBR | ND | NA | |
| VF20/120B | 26/8/2020 | M/18 | 2+ paired CB or short chains; 3+ PBR | ND | *Clostridium* sp. | KC331191.1 |
| VF20/120C | 26/8/2020 | M/8 | 3+ paired CB or short chains; PBR | ND | NA | |
| VF20/120D | 26/8/2020 | F/15 | 1+ paired CB; 3+ PBR; ALO | ND | *Clostridium* sp. | FJ384387.1 |
| VF20/120E | 26/8/2020 | F/1.5 | 5+ PBR | ND | NA | |
| VF20/124 | 1/9/2020 | M/10 | Few short bipolar CB; 4+ PBR | ND | *Clostridium vulturis* | NR_148266.1 |
| VF20/129 | 7/9/2020 | M/18 | 2+ bipolar short rods, paired CB | *P. multocida*[d], *Klebsiella pneumoniae*[c] | Mixed culture signal | |
| VF20/130 | 12/9/2020 | M/4 | 2+ bipolar short rods, paired CB | ND | *P. multocida* | LR134514.1 |
| | | | | | Bisgaard taxon 45 | AY683487.1 |
| VF20/134 | 16/9/2020 | F/18 | Paired CB; 3+ PBR | ND | *Clostridium* sp. | AY685918.1 |
| VF20/170 | 9/11/2020 | M/30 | 1+ bipolar short rods, paired CB | No *Pasteurella* spp.[b] | Bisgaard taxon 45 | AY683487.1 |

*CB* coccobacilli, *ALO Anaplasma*-like organisms, *PBR* putrefactive bacterial rods, *ND* not done, *NA* no amplification.
a = closest match from NCBI database.
b = result from ZimVet.
c = result from VetDiagnostix.
d = result from Zimbabwe Central Veterinary Laboratories.
* = intracellular bacteria observed.

mortality event in saiga was associated with high temperatures and humidity along with stresses due to calving[28]. We propose that stress from a combination of heat, drought, and population density were likely contributing factors in this outbreak. Food and water resources normally wane as temperatures rise during the dry season, and elephants must travel increasing distances between water points and foraging areas[29,30]. Matabeleland North, the province where the mortalities occurred, is prone to drought events[31], and Zimbabwe as a whole experienced very poor rains in the 2018/2019 wet season (November–March) and the first half of the 2019/2020 wet season[32]. Although there was not culture or molecular evidence to confirm Bisgaard taxon 45 in more than six mortalities in Zimbabwe, the elephants examined were in good body condition and unlikely to have died of drought-related starvation or severe dehydration alone. Local elephant densities are high; the KAZA Elephant Survey in the 2022 dry season estimated a population of 61,531 elephants in north–west Matabeleland, with a density of 2.457/km[25],. The resulting high demand for food in combination with the existing rainfall deficit would have compounded on the expected natural resource scarcities in the dry season. A similar multifactorial explanation could underlie the mortality event in Botswana, where elephants experienced stress from poaching in preceding years[11] and high density (2.31 elephants/km$^2$) due to veterinary fencing that restricts dispersal[10,11,33], as well as a dry 2019[10].

The source of infection and route of transmission remain unknown in this outbreak. Bisgaard taxon 45 has been isolated from clinically healthy psittacines[17] and may represent a previously unknown part of elephants' normal flora in this region. Intraspecific transmission via inhalation or ingestion is possible, especially given the highly social behaviour of elephants, which includes contact between trunks or even placing trunks in each others' mouths. When hot and stressed, elephants have been observed withdrawing fluid from a unique pharyngeal pouch and spraying this over their bodies[34], and such a pouch might be colonized by Bisgaard taxon 45. *P. multocida* is known to multiply rapidly in carcasses of animals with haemorrhagic septicaemia[27], and

the natural curiosity of elephants about their dead could serve as an additional opportunity for exposure. There is no known health risk from affected carcasses to natural scavengers such as vultures and hyenas. No elephants had evidence of large cat bite wounds, implicated in prior human infections[14]. Bisgaard taxon 45's zoonotic potential via other routes of transmission is unknown. Human infections with other *Pasteurella* spp. occur through animal contact, including bites, scratches, licking wounds, or contact with mucous secretions[35].

More research is needed to understand the epidemiology of Bisgaard taxon 45 in elephants and whether latent carriers or other species play a role in maintaining this organism. Bacterial septicaemia adds to a growing list of disease-related threats to elephant conservation, including tuberculosis[36], anthrax, elephant endotheliotropic herpesvirus, encephalomyocarditis virus, floppy trunk syndrome, and malicious poisoning[37]. It should be considered as a differential diagnosis in sudden mortality of African elephants, and there is a need to simplify permit processes to facilitate timely sampling and testing of animals in mortality events. The epidemiologic associations of bacterial septicaemia with extreme weather events and temperature stress may make outbreaks more likely under predicted climate change, which includes warming temperatures, more intense precipitation events, increased humidity, or increased drought[38]. Unfortunately, the high population density of elephants in north-western Zimbabwe makes them more susceptible to pressures from drought or other environmental events leading to resource depletion. Establishing wildlife corridors and maintaining habitat connectivity within the KAZA TFCA to allow elephant dispersal is crucial to the long-term conservation of the species[11].

## Methods
### Ethical statement
Ethical approval was not required as no live animals were involved in this study. Victoria Falls Wildlife Trust (VFWT) operates a registered veterinary diagnostic laboratory and collected samples on behalf of the Zimbabwe Parks and Management Wildlife Authority (ZPMWA).

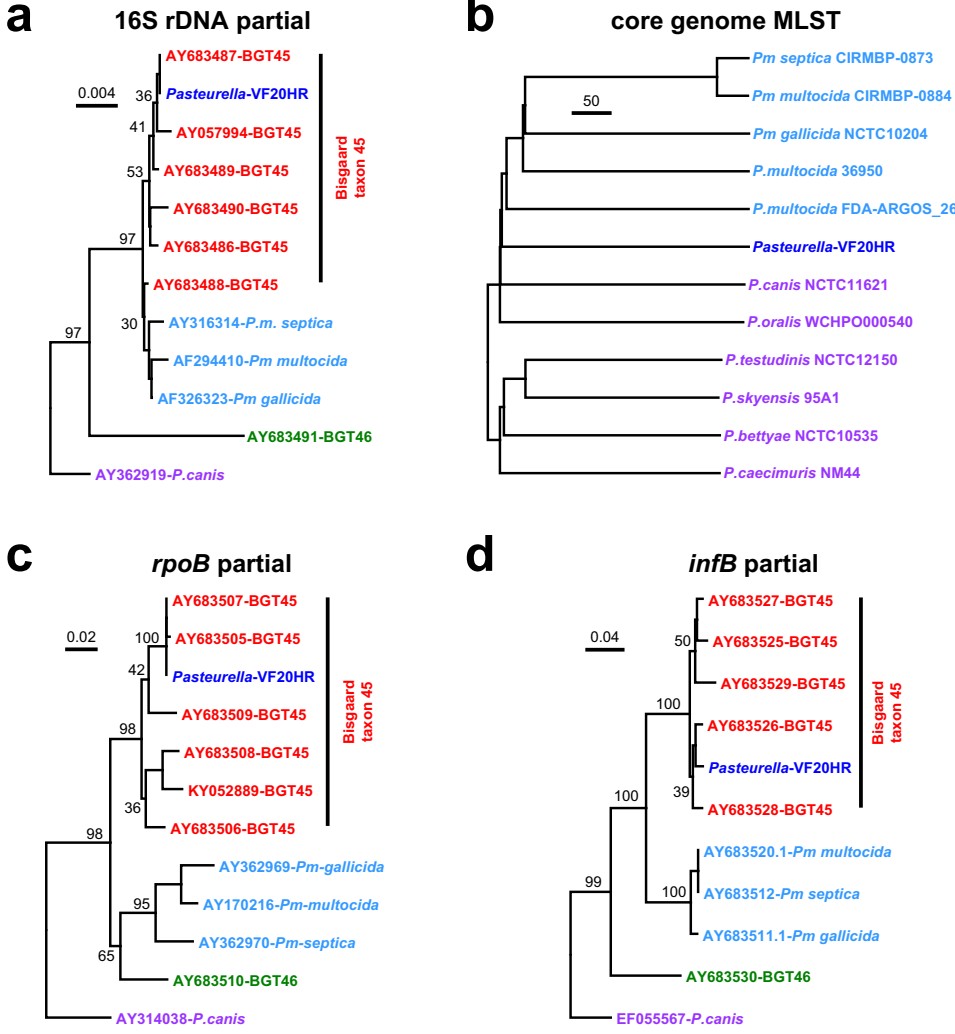

**Fig. 4 | The VF20HR isolate is a member of the Bisgaard Taxon 45 (BGT45) family.** The phylogenetic relationship of VF20HR with other members of the *Pasteurellaceae* family was analysed using (**a**) 16 S rDNA sequences (1177 nucleotides); (**b**) core genome multilocus sequence typing (MLST) based on 681 markers; (**c**) partial *rpoB* gene sequences (467 nucleotides) and (**d**) partial *infB* gene sequences (455 nucleotides). For the 16 S rDNA, *rpoB* and *infB* trees, the Genbank accession numbers for the sequences from Bisgaard Taxon 45 (BGT45), Bisgaard Taxon 46 (BGT46), *Pastereulla multocida* subspecies (*Pm*) and *Pasteurella canis* are provided in the tree. The support for monophyletic groups by bootstrap analysis is indicated as percentages. Samples are color-coded, with dark blue (VF20HR), red (Bisgaard Taxon 45), green (Bisgaard Taxon 46), light blue (*P. multocida* subspecies) and purple (*P. canis*).

The Memorandum of Understanding between VFWT and ZPMWA to support conservation activities in the Victoria Falls area of Zimbabwe includes wildlife veterinary sampling, and elephants were sampled opportunistically as part of routine postmortem investigations conducted by VFWT.

## Veterinary investigation

Elephant mortalities were reported to the VFWT over the course of several weeks, starting on 24 August 2020 at locality S 18.11204, E 25.64996. Aerial surveys were conducted on 3–5 September (11 h 35 min flying time, ~1440 km$^2$) and 19–20 September 2020 (8 h 30 min flying time, ~615 km$^2$) to search a larger area for carcasses. Of 35 carcasses identified either from the air, or located on the ground, 25 were visited on foot. However, only 15 of these were suitable for sampling to a varying degree, the other 10 being too decomposed. Only partial post-mortem examinations were conducted as anthrax was initially suspected and could not be ruled out, based on the gross appearance of the carcasses and organs. Five carcasses were opened on the left side exposing the abdominal and thoracic viscera for the collection of organ samples and, in one carcass, the brain was also collected.

Skeletal muscle or tongue were the only tissues collected from the other 10 carcasses, while blood and blood smears were taken from all 15. Swabs in charcoal agar for bacterial culture were collected directly from 5 carcasses. Tissues were both collected fresh and placed in 10% buffered formalin (BF), blood was collected for blood smears, and fresh stomach contents were collected for cyanide testing, using a simplified alkaline picrate strip test[39]. In all cases samples were collected into new containers or individually wrapped swabs labelled with the elephant's identification number, and post-mortem instruments were washed and disinfected with F10 (quaternary ammonium compounds and biguanide, Health and Hygiene (Pty) Ltd, Roodepoort, South Africa) and/or sodium hypochlorite between post-mortem examinations to avoid cross-contamination. Details of further veterinary investigations are provided in the Supplementary Methods.

## Histopathology

Tissue samples were fixed in 10% BF at the post-mortem examination at VFWT and Vetdiagnostix. Tissue samples from elephants VF20/112 (spleen, heart, lung, liver, stomach and cerebral cortex) and VF20/113

(spleen, heart, liver, lymph node and kidney) were also sent for further examination, after the formalin was drained off and alcohol moistener added. Samples were re-immersed in 10% BF for a 72-h period before being processed for histopathology. Serial 4 μm thick sections were cut and tissue sections of multiple organs, including liver spleen, kidney, lung, brain and muscle were stained with hematoxylin and eosin (H&E), or Gram Twort for bacterial colony tinctorial visualization.

## Microbiological and genomic diagnostics
Samples showing Gram-negative, short coccobacillary bacteria were used for microbiological and molecular investigations to identify the potential causative agents. A more detailed description of methods can be found in the Supplementary Methods.

**Microbiological investigation.** The brain, liver and spleen samples from elephant VF20/112 were cultured aerobically on 5% sheep blood tryptose agar, cross-streaked with a V factor producing *Staphylococcus aureus* and incubated at 35°C in an atmosphere of 5–10% $CO_2$. The samples were also cultured aerobically on MacConkey agar without crystal violet, nutrient broth, Rappaport Vassiliadis (RV) broth for *Salmonella* enrichment and CHROMagarTM *Pasteurella* agar (CHROMagar, Paris, France) at 35°C. The RV broth was sub-cultured after 24 h onto XLD (Xylose-Lysine-Desoxycholate) agar to detect *Salmonella*. Bacterial isolates were phenotypically identified according to standard methods[40].

**16S rDNA typing.** To determine the nature of bacteria detected, DNA was extracted using a Quick-DNA™ Fungal/Bacterial Miniprep Kit (Zymo Research, Irvine, CA, USA) and the 16S rDNA target region amplified with primers 27 F (5' AGAGTTTGATCMTGGCTCAG 3') and 1492 R (5' CGGTTCFTTTFTTACFACTT 3')[41–43]. Products were analysed on an agarose gel and sequenced using an ABI 3500XL sequencer, and further analysed using MEGA 7 alignment and phylogenetic software[44].

**Whole genome sequencing.** To characterise the bacterium further, high-throughput sequencing (HTS) using Illumina MiSeq was carried out on DNA extracted directly from liver and spleen samples. Due to the overrepresentation of *Loxodonta africana* sequences, the sequencing reads had to be mapped first against the *L. africana* genome sequence (version LoxAfr3, accession number GCA_000001905.1) using Bowtie2 2.4.2[45] and Samtools 1.12[46] as described[47], and reads mapping to the *L. africana* genome were removed. Initial assembly with the remaining reads using Shovill version 1.1.0 (https://github.com/tseemann/shovill) and the Spades assembler version 1.13.3 with a coverage cut-off of 10, gave a 2.5 Mbp assembly consisting of 264 contigs. Further work using selective sequence coverage and comparison with genome sequences of *Pasteurellaceae* spp. allowed removal of most of the small contigs, leaving a final genome assembly consisting of 14 contigs totalling 2,240,110 nt, with assembly metrics of $L_{50} = 2$ and $N_{50} = 511,117$. This genome assembly was used for further investigations.

**Comparative genomics.** A total of 279 *Pasteurellaceae* genome sequences were downloaded from the NCBI Genome section using NCBI-genome-download version 0.3.1 (https://github.com/kblin/ncbi-genome-download) (Supplementary Data 1). Genomes were subsequently annotated using Prokka version 1.14[48]. The presence of *Pasteurella* virulence markers was based on screening using BLAST+ with gene sequences and amino acid translations obtained from Genbank, based on the list provided in a recent review on *Pasteurella* virulence[15]. Comparisons with the Bisgaard 45 taxon were based on the 16S rDNA, *infB* and *rpoB* sequences of Bisgaard taxons isolated from large-cat bite wounds and psittacine birds, using MEGA7[44].

## Reporting summary
Further information on research design is available in the Nature Portfolio Reporting Summary linked to this article.

## Data availability
The Illumina sequencing reads and genome assembly of Bisgaard Taxon 45 isolate VF20HR generated in this study have been deposited in the SRA and NCBI Genome repositories, under SRA accession code SRR22922729, Bioproject accession code PRJNA914783, Biosample accession code SAMN32358138 and GenBank accession number JAQAHH000000000. The *Pasteurella* cgMLST scheme is available via Figshare (https://doi.org/10.6084/m9.figshare.21791843.v1).

## Code availability
No code was generated in the current study.

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

## Acknowledgements

We thank VWFT staff for data collection, surveying for carcasses, and laboratory sample processing. We thank pilot Ed Whitfield (Flying for Wildlife Zimbabwe) for conducting aerial surveys. We thank Dr Kudzai Mupondi and Zimbabwe National Parks rangers who assisted with locating carcasses, postmortem examinations, and carcass disposal, and Greg Foggin for assistance with postmortems. We thank Drs Rick Last and Steven Kubiski for consultation on histopathologic findings. Jos Danckwerts and Wild is Life Trust management and staff are acknowledged for providing much assistance in locating and processing carcasses, as well as providing helicopter time. We thank San Diego Zoo Global and Oak Foundation for general project support. We thank Steve Alexander and Rob Rees for sample submissions from prior elephant mortality events. The Director General has given permission for the publication of this paper dealing with a CITES listed species under the jurisdiction of the Zimbabwe Parks and Wildlife Management Authority. General funding for diagnostic testing and laboratory support at VFWT was provided by USAID. Vetdiagnostix funded sample cultures and permit acquisition in South Africa. Design Biologix funded the preliminary molecular research done on the organism. Sample submission and laboratory work in the UK was funded by Defra (new & emerging diseases, NED) and the FCDO UK in a joint effort in support of wildlife health.

## Author contributions

C.M.F. and L.E.R. conceived the initial write-up. C.M.F. conducted all field work. C.C. and H.C.R. provided resources, carried out project administration and facilitated issuance of permits. L.E.R. compiled and summarized data, produced tables and maps, and drafted the original manuscript (with contributions from C.M.F.). L.E.R., T.F., and A.N. produced figures. C.M.F., M.M.H., A.B., T.F., A.D.T., J.T., A.S.L., R.J.E., A.D.,

A.N., A.H.M.v.V., and F.S. carried out analyses and interpretation of results. All authors contributed to subsequent drafts and revisions of the paper, led by F.S. and L.E.R.

## Competing interests

Design Biologix has funded the molecular research done on the organism. Dr Angela Buys is employed by Design Biologix (full-time) and acted in her capacity as researcher. M.M.H. is an employee of Vetdiagnostix, a commercial veterinary diagnostic laboratory, which funded sample cultures and permit acquisition. The other authors declare no competing interests.
