## [Peer Review File · Nature Communications]

REVIEWER COMMENTS

Reviewer #1 (Remarks to the Author):

Foggin et al, investigated comprehensively the cause(s) of mortality of 35 elephants in Zimbabwe. Authors would like to connect this mortality to an extensive outbreak occurred in Botswana, but it seems that the Bisgaard taxon 45 was not identified in those elephants in Botswana. The authors mentioned that in the hot dry season of 2019, at least 200 elephants in the area died as a result of drought and starvation. These 35 elephants also died over the course of late August to November 2020, the hottest time in Zimbabwe. Therefore, it looks that main cause of the death could be drought and starvation, and secondary to this situation some bacterial septicemia happened and maybe this is the reason that authors used the word secondary in line 78.

Although I like the study and I think it has high value to be published, but there are several weak points. Maybe the biggest weak point of the study is lack of cultivation and isolation data. It is unbelievable that cultivation was not attempted for majority of samples. If there are still some samples somewhere available, I highly recommend analyzing them. As I understood because of anthrax, post-mortem examination was not done regularly and samples for histopathology and/or culture were not collected.

Unfortunately, many wild animals mortality investigation missions, miss even the primary principles of good laboratory practice, such as complete cleaning of post-mortem examination tools such as knives and scissors; and sample collection with contaminated and dirty hands and instruments. Please explain comprehensively how the field investigators avoided cross contamination between the samples (if they have done it).

For sure, the authors also agree that only one cultivation positive sample (based on Table 1) cannot be a sophisticated indication that Bisgaard taxon 45 was the causative agent of the outbreak. I agree the genome sequencing data gave us evidence, but we should consider the high chance of cross contamination in this case, as most genome sequencing positive samples belong to 24th and 25th August; In addition there are other examples, such as sample VF20/170 with no *Pasteurella*-like bacteria growing caused the Bisgard signal.

Therefore, please show more evidence that the Bisgaard taxon 45 may act as a primary cause of the mortality (and it looks that this is the conclusion that you prefer); or change the title and abstract, to somehow emphasize the role of the bacteria as a secondary and opportunistic infection (you already mentioned that the bacteria is a commensal bacteria).

Regarding the figure 3. The Magnifications should be indicated for each image (or once in the legend if all of them have the same magnification). Only for three images, higher magnifications of bacterial colonies or clusters (in small squares) were provided, and two of the, including the liver (right image) and lymph node (right image) have relatively acceptable sharpness and magnification. Please provide sharp images of bacterial colonies with high magnifications for all of them.

Reviewer #2 (Remarks to the Author):

Review

Pasteurella sp. causes fatal septicaemia in African elephants in Zimbabwe

This manuscript presents important data regarding a cluster of 35 elephant mortalities in North Western Zimbabwe in 2020. Unusual numbers of elephants also died in neighbouring Botswana in the same year, the cause of which has not been definitively determined due to the difficulties in obtaining optimum samples from the remote area where these elephants died. This manuscript provides evidence that bacterial infection may have been the cause of the elephant deaths in Zimbabwe. Given the steady population decline due to poaching by organised crime syndicates and habitat fragmentation of this iconic African megaherbivore, the data presented are valuable to ecologists, landscape managers and veterinarians in Africa.

The authors detected a bacterium related to *Pasteurella multocida* in 7 of 15 dead elephants; and describe pathological evidence of bacterial infection in 5 elephants. However, the text and Table 1 do not clearly correlate the summarized pathology findings with the culture and molecular diagnoses for each animal. Without this, the causal relationship between the presence of the bacterium and the associated lesion cannot be evaluated. A clear explanation, using tables and animal identification numbers, of the number of carcasses found, number in which necropsy examination could be done, number in which histological examination was done, degree of post-mortem change and pathological findings in each case (as well as the date of death, age, sex, blood smear, culture, 16S rDNA results) is needed to better illustrate the strength of the association between the presence of the bacterium and the cause of death. Which labs did which tests on which samples from which animal and the results obtained should also be clearly set out. Not all the information needs to be in the main part of the manuscript but the correlation between the key pathology findings and bacterial results is necessary. This would greatly clarify and justify the conclusions reached.

In Table 1, were any of the bacteria seen in the blood smears intracellular (indicating phagocytosis before death)?

The wording needs to be more concise throughout, replacing non-specific terms such as 'leucocytic', 'vascular structures', 'generally', 'usually', 'several', 'frequently' with exact numbers of animals, samples, tests, post-mortem examinations done in all portions of the manuscript.

The meaning of terms such as 'dry' or 'wet' season needs to be outlined for readers outside of southern Africa.

In the discussion, the evidence for septicaemia and the putative cause in these cases should be separated from the discussion regarding *Pasteurella multocida* and possible risk factors for septicaemia in elephants. The discussion would be improved by addressing suggestions that the mortality event in Botswana was associated with high elephant densities (RM Huang et al., 2022; RJ van Aarde et al., 2021; S Azeem et al., 2020), and how this might relate to these reported cases from Zimbabwe.

The caption for Fig 2 should indicate from which animal the images were taken and whether or not the vascular lesions were passive (congestion), active (hyperaemia) or haemorrhage.

Figure 2: Images A and C are similar. Image A would be better as an inset for Image C.

Figure 3: the asterisks obscure smaller lesions and the same symbol is used for different lesions: necrosis, microabscessation, and thrombi. The lymph node images are not necessary as the lesions are similar to other organs. The labelling of each image in Figure 2 should be alphabetical and the caption indicate from which animal the tissue was taken.

In my opinion, the evidence as provided is strongly suggestive but needs to be more clearly presented. The information is original and of considerable interest and value to the management of elephants.

I am not able to comment on the technical aspects of the microbiological, toxicological or molecular genetic analyses. There is no inflammatory language used, or inappropriate handling of gender issues.

We thank the reviewers for the insights and suggestions, and we have responded to the reviewers' comments below and made suggested changes to the manuscript. Below in plain text are our responses to individual reviewer comments. Line numbers refer to the revised manuscript:

REVIEWER COMMENTS

Reviewer #1 (Remarks to the Author):

Foggin et al, investigated comprehensively the cause(s) of mortality of 35 elephants in Zimbabwe. Authors would like to connect this mortality to an extensive outbreak occurred in Botswana, but it seems that the Bisgaard taxon 45 was not identified in those elephants in Botswana. The authors mentioned that in the hot dry season of 2019, at least 200 elephants in the area died as a result of drought and starvation. These 35 elephants also died over the course of late August to November 2020, the hottest time in Zimbabwe. Therefore, it looks that main cause of the death could be drought and starvation, and secondary to this situation some bacterial septicemia happened and maybe this is the reason that authors used the word secondary in line 78.

Although I like the study and I think it has high value to be published, but there are several weak points. Maybe the biggest weak point of the study is lack of cultivation and isolation data. It is unbelievable that cultivation was not attempted for majority of samples. If there are still some samples somewhere available, I highly recommend analyzing them. As I understood because of anthrax, post-mortem examination was not done regularly and samples for histopathology and/or culture were not collected.

We agree that more culture data would be ideal. We were limited by the fact that decomposition set in rapidly during the hot dry season and samples were degraded. Bacteriological culture facilities in Zimbabwe have limited capacity and their quality is uncertain. While it would have been preferable to send more samples to South Africa, obtaining necessary permits in a timely manner to facilitate additional bacterial cultures was not possible, as outlined in the discussion. It took 32 days to acquire permits to send samples to South Africa and the positive sample was cultured over 2 months after collection. Studies of *Pasteurella multocida* and related species show very variable results for survival in transport medium, which is dependent on many factors, not the least being the initial level of the pathogen and the number of contaminants in the sampled tissues. We have added lines 206-207 to further address the constraints on culturing.

We did, however, manage to isolate Bisgaard taxon 45 once. We further believe that the detection by qPCR, in combination with the histological findings of septicaemia and bacteria in blood smears provides strong evidence for Bisgaard taxon 45 being a cause of the septicaemia, which we found in 13/15 cases.

Unfortunately, many wild animals mortality investigation missions, miss even the primary principles of good laboratory practice, such as complete cleaning of post-mortem examination tools such as knives and scissors; and sample collection with contaminated and dirty hands and instruments. Please

explain comprehensively how the field investigators avoided cross contamination between the samples (if they have done it).

Post-mortem instruments were washed and disinfected with QAC and/or bleach in between post-mortem examinations. All samples were collected into new containers or with individually wrapped swabs in the field. Samples were clearly labelled with each animal's identification number. We have added lines 474-478 to clarify these procedures. In the lab, standard operating procedures for PCR were followed to avoid contamination.

While contamination could possibly explain PCR positive results, it is not an explanation for the septicaemia and bacteria in smears detected in 13/15 samples.

For sure, the authors also agree that only one cultivation positive sample (based on Table 1) cannot be a sophisticated indication that Bisgaard taxon 45 was the causative agent of the outbreak. I agree the genome sequencing data gave us evidence, but we should consider the high chance of cross contamination in this case, as most genome sequencing positive samples belong to 24th and 25th August; In addition there are other examples, such as sample VF20/170 with no Pasteurella-like bacteria growing caused the Bisgard signal.

Therefore, please show more evidence that the Bisgaard taxon 45 may act as a primary cause of the mortality (and it looks that this is the conclusion that you prefer); or change the title and abstract, to somehow emphasize the role of the bacteria as a secondary and opportunistic infection (you already mentioned that the bacteria is a commensal bacteria).

The pathological evidence indicates that this was a bacterial septicaemia but there may have been other factors involved, as we outlined in the discussion. It is not clear whether Bisgaard taxon 45 is a facultatively pathogenic bacterium in elephants that otherwise occurs as commensal. In the latter case, however, its presence would be restricted to mucosal surfaces, not internal organs where we detected it by PCR. We have revised the title, abstract, introduction, and discussion (lines 38, 47, 81, and 166) to reflect the association between Bisgaard taxon 45 and the mortalities.

Regarding the figure 3. The Magnifications should be indicated for each image (or once in the legend if all of them have the same magnification). Only for three images, higher magnifications of bacterial colonies or clusters (in small squares) were provided, and two of the, including the liver (right image) and lymph node (right image) have relatively acceptable sharpness and magnification. Please provide sharp images of bacterial colonies with high magnifications for all of them.

We have revised Figure 3 by reducing the number of images. The magnifications, stains, and animal IDs are now included in the legend (lines 428-436). We have included one image of Gram-negative bacteria to avoid repetition and provided images of necrosuppurative foci and Gram-negative coccobacilli to represent the pathology observed.

Reviewer #2 (Remarks to the Author):

Review

Pasteurella sp. causes fatal septicaemia in African elephants in Zimbabwe

This manuscript presents important data regarding a cluster of 35 elephant mortalities in North Western Zimbabwe in 2020. Unusual numbers of elephants also died in neighbouring Botswana in the same year, the cause of which has not been definitively determined due to the difficulties in obtaining optimum samples from the remote area where these elephants died. This manuscript provides evidence that bacterial infection may have been the cause of the elephant deaths in Zimbabwe. Given the steady population decline due to poaching by organised crime syndicates and habitat fragmentation of this iconic African megaherbivore, the data presented are valuable to ecologists, landscape managers and veterinarians in Africa.

The authors detected a bacterium related to *Pasteurella multocida* in 7 of 15 dead elephants; and describe pathological evidence of bacterial infection in 5 elephants. However, the text and Table 1 do not clearly correlate the summarized pathology findings with the culture and molecular diagnoses for each animal. Without this, the causal relationship between the presence of the bacterium and the associated lesion cannot be evaluated. A clear explanation, using tables and animal identification numbers, of the number of carcasses found, number in which necropsy examination could be done, number in which histological examination was done, degree of post-mortem change and pathological findings in each case (as well as the date of death, age, sex, blood smear, culture, 16S rDNA results) is needed to better illustrate the strength of the association between the presence of the bacterium and the cause of death. Which labs did which tests on which samples from which animal and the results obtained should also be clearly set out. Not all the information needs to be in the main part of the manuscript but the correlation between the key pathology findings and bacterial results is necessary. This would greatly clarify and justify the conclusions reached.

We have added two tables to the supplemental information for clarity, and these are referenced in the manuscript on lines 105-107. Supplemental Table 1 addresses the gross pathology and histopathology. Supplemental Table 2 addresses other tests performed at various labs and the results.

In Table 1, were any of the bacteria seen in the blood smears intracellular (indicating phagocytosis before death)?

Yes, there were intracellular bacteria observed. We have clarified this in the text (line 121).

The wording needs to be more concise throughout, replacing non-specific terms such as 'leucocytic', 'vascular structures', 'generally', 'usually', 'several', 'frequently' with exact numbers of animals, samples, tests, post-mortem examinations done in all portions of the manuscript.

We have revised and removed imprecise wording throughout the manuscript and supplemental information (lines 107, 110, 113-114, 116 in manuscript and lines 167, 169, 171, 184, 190, 192, 199, 201, 202, 203, 206-207, 214, 217, and 219 in the supplemental information).

The meaning of terms such as 'dry' or 'wet' season needs to be outlined for readers outside of southern Africa.

We have added parenthetical statements on lines 70, 72, and 231 that include the corresponding months where reference is made to dry or wet seasons.

In the discussion, the evidence for septicaemia and the putative cause in these cases should be separated from the discussion regarding *Pasteurella multocida* and possible risk factors for septicaemia in elephants.

We have added to the discussion on the evidence from our samples (lines 164-166) and reordered part of the discussion (lines 211-220) to improve the flow from discussion of the evidence to the discussion on *P. multocida* and risk factors.

The discussion would be improved by addressing suggestions that the mortality event in Botswana was associated with high elephant densities (RM Huang et al., 2022; RJ van Aarde et al., 2021; S Azeem et al., 2020), and how this might relate to these reported cases from Zimbabwe.

We have added lines 235-237 noting similarities in rainfall, possible stressors in the elephant population in Botswana and the high densities associated with veterinary fencing in the affected area, but do not want to make any definitive comments on the outbreak in Botswana given that no samples from Botswana were included here. We have also added RM Huang et al., 2022 to the references (the other papers listed had already been cited).

The caption for Fig 2 should indicate from which animal the images were taken and whether or not the vascular lesions were passive (congestion), active (hyperaemia) or haemorrhage.

The caption has been updated with the corresponding animal ID number and updated terminology for the vascular lesions (lines 419-424).

Figure 2: Images A and C are similar. Image A would be better as an inset for Image C.

We have left the images separate as they originated from different animals, which is now noted in the caption.

Figure 3: the asterisks obscure smaller lesions and the same symbol is used for different lesions: necrosis, microabscessation, and thrombi. The lymph node images are not necessary as the lesions are similar to other organs. The labelling of each image in Figure 2 should be alphabetical and the caption indicate from which animal the tissue was taken.

Figure 3 and its captions have been revised to more clearly indicate the lesions in each image (lines 428-436). Alphabetical labels and animal IDs have been added.

In my opinion, the evidence as provided is strongly suggestive but needs to be more clearly presented. The information is original and of considerable interest and value to the management of elephants.

We thank the reviewer for the interest in our study and the constructive comments to improve the manuscript. We hope the reviewer agrees that the review process has led to a clearer presentation as requested.

I am not able to comment on the technical aspects of the microbiological, toxicological or molecular genetic analyses. There is no inflammatory language used, or inappropriate handling of gender issues.

REVIEWER COMMENTS

Reviewer #1 (Remarks to the Author):

The authors addressed all of my queries.

Thank you

Reviewer #2 (Remarks to the Author):

Pasteurella sp. associated with fatal septicaemia in African elephants in Zimbabwe

The changes made to the manuscript document more clearly:

1. the presence of septicaemia in 4 of the 5 elephants on which necropsy examinations could be performed. The inflammatory changes in the liver and lung of the 5th elephant (VF20/129) were not fully described and the diagnosis of septicaemia was not made in this animal. Different bacterial culture results were obtained from the blood (*Pasteurella multocida*) and a lung swab (*Klebsiella pneumoniae*) from this elephant.
2. the presence of Bisgaard taxon 45 in samples from 6 elephants, including the 4 with septicaemia, and 2 additional elephants on which necropsy examinations could not be performed. In 4 of these elephants, samples from several organs or blood were positive, supporting the likelihood of septicaemia. In one elephant a mixed result (with *P. multocida*) was obtained.

I feel that the reorganised evidence supports the conclusions of the authors for these 6 animals. In the interests of clarity, changes to the title and text should be made to reflect this:

1. The number 6 should be added to the title
2. And to line 42 of the abstract
3. Line 42 should also state that detailed genomic analysis was only one in one animal.
4. It should be clear that only 1 elephant brain was sampled (lines 173 and 203 in the supplementary material).

5. Line 184 in the manuscript should be modified to indicate that the statement refers to only 2 elephant brains (1 from this series, and a previous case).

6. Extrapolation to the other 29 animals is less certain (given the described role of drought in line 213 of the manuscript), and should be briefly addressed in the discussion. The culture and molecular results from VF20/129 should be discussed.

The text mentions haemorrhages in organs but does not state which organs were affected. Were these the ones with necrotic foci/bacterial emboli? The only ones illustrated were in the spleen. This should be clarified.

The evidence for human infections from skin wounds etc is presented twice in one paragraph (lines 226 and 238) and should be combined. The absence of cutaneous wounds or predator bites in the elephants is important, however, in terms of the source of the infections.

Since intracellular phagocytosed bacteria are important to distinguish ante and post-mortem bacteraemia, I suggest that Table 1 is modified (perhaps by a star or bold type) to show in which animals the blood smear showed intracellular bacteria.

The additional tables are very helpful in evaluating the data presented.

The wording is much more concise.

The reordered discussion and figures are much improved.

Is there perhaps an aerial or close up photograph of one of the elephants before necropsy, showing the degree of autolysis (or lack of it) to replace one of the pictures of the spleen in Figure 2?

NCOMMS-23-05144B, Response to Reviewers' comments

We thank the reviewers for the insights and suggestions, and we have responded to the reviewers' comments below and made suggested changes to the manuscript. Below in plain text are our responses to individual reviewer comments. Line numbers refer to the revised manuscript:

Reviewer #1 (Remarks to the Author):

The authors addressed all of my queries.

Thank you

We thank Reviewer #1 again for their comments and are glad to have addressed all their queries.

Reviewer #2 (Remarks to the Author):

Pasteurella sp. associated with fatal septicaemia in African elephants in Zimbabwe

The changes made to the manuscript document more clearly:

- 1. the presence of septicaemia in 4 of the 5 elephants on which necropsy examinations could be performed. The inflammatory changes in the liver and lung of the 5th elephant (VF20/129) were not fully described and the diagnosis of septicaemia was not made in this animal. Different bacterial culture results were obtained from the blood (*Pasteurella multocida*) and a lung swab (*Klebsiella pneumoniae*) from this elephant.**
- 2. the presence of Bisgaard taxon 45 in samples from 6 elephants, including the 4 with septicaemia, and 2 additional elephants on which necropsy examinations could not be performed. In 4 of these elephants, samples from several organs or blood were positive, supporting the likelihood of septicaemia. In one elephant a mixed result (with *P. multocida*) was obtained.**

I feel that the reorganised evidence supports the conclusions of the authors for these 6 animals. In the interests of clarity, changes to the title and text should be made to reflect this:

- 1. The number 6 should be added to the title**

We have changed the title as suggested.

- 2. And to line 42 of the abstract**

- 3. Line 42 should also state that detailed genomic analysis was only one in one animal.**

We have altered lines 41-42 as follows:

Post-mortem and histological findings suggested a bacterial septicaemia similar to haemorrhagic septicaemia caused by *P. multocida*. Biochemical tests and 16S rDNA analysis of six samples and genomic analysis of one sample confirmed the presence of Bisgaard taxon 45.

- 4. It should be clear that only 1 elephant brain was sampled (lines 173 and 203 in the supplementary material).**

We have added the word “one” to lines 175 and 204 of the supplementary material.

5. Line 184 in the manuscript should be modified to indicate that the statement refers to only 2 elephant brains (1 from this series, and a previous case).

We have added the word “two” to line 186.

6. Extrapolation to the other 29 animals is less certain (given the described role of drought in line 213 of the manuscript), and should be briefly addressed in the discussion. The culture and molecular results from VF20/129 should be discussed.

We have added the following sentence to lines 237-239 in the discussion to address the lack of confirmation from all elephants in this mortality event:

Although there was not culture or molecular evidence to confirm Bisgaard taxon 45 in all 35 mortalities in Zimbabwe, the elephants examined were in good body condition and unlikely to have died of starvation or severe dehydration due to drought alone.

We have added interpretation of the VF20/129 results to the discussion on lines 196-210:

Such delays preclude the use of time-sensitive assays such as bacterial culture and postpone follow-up diagnostic testing indicated by initial results. As an example, samples from elephant VF20/129 were first sent to an in-country laboratory for culture, which did not require additional time to obtain import and export permits. The *P. multocida* result from this laboratory was likely Bisgaard taxon 45, which would have been indistinguishable from *P. multocida* when serotyping or biochemical tests were not attempted. It took 32 days to acquire permits to send samples to South Africa, and samples were only received at the laboratory eight weeks after collection, which is the likely reason for the failure to isolate Bisgaard taxon 45 there. Only *Klebsiella pneumoniae* was isolated from VF20/129 at this laboratory, where a single colony was isolated from the lung sample and on one of three initial media on which it was inoculated. This suggests that there were few *Klebsiella* bacteria present. The *Klebsiella* finding was likely incidental; *Klebsiella* are part of the normal flora and were isolated from two other unrelated elephant samples sent at the same time, all with different colony characteristics and antibiotic resistance profiles. The 16S RNA results support the presence of a mixed culture, presumed to be Bisgaard taxon 45 and *Klebsiella pneumoniae*.

The text mentions haemorrhages in organs but does not state which organs were affected. Were these the ones with necrotic foci/bacterial emboli? The only ones illustrated were in the spleen. This should be clarified.

Yes, haemorrhages were observed in organs with bacterial emboli. We have added to lines 107-108 in the main text to clarify which organs were affected:

Briefly, the carcasses were in average body condition with hepatomegaly and splenomegaly as the most prominent gross pathological findings, with variable haemorrhages (Fig. 2) across the epicardium, liver, lungs, intestinal serosae, hepatic and splenic lymph nodes, and in one case, the diaphragm.

We have also added to lines 169-170 in the supplemental information:

Although haemorrhages were present, they varied in intensity and distribution (Fig. 2); across the epicardium, liver, lungs, intestinal serosae, hepatic and splenic lymph nodes, and in one case, the diaphragm.

We also added two panels to Figure 2 to illustrate haemorrhages in other organs.

The evidence for human infections from skin wounds etc is presented twice in one paragraph (lines 226 and 238) and should be combined. The absence of cutaneous wounds or predator bites in the elephants is important, however, in terms of the source of the infections.

We have altered lines 257-259 as follows:

No elephants had evidence of large cat bite wounds, implicated in prior human infections¹⁴. Bisgaard taxon 45's zoonotic potential via other routes of transmission is unknown.

Since intracellular phagocytosed bacteria are important to distinguish ante and post-mortem bacteraemia, I suggest that Table 1 is modified (perhaps by a star or bold type) to show in which animals the blood smear showed intracellular bacteria.

An asterisk has been added to Table 1 to denote where intracellular bacteria were observed in the slides available for review.

The additional tables are very helpful in evaluating the data presented.

The wording is much more concise.

The reordered discussion and figures are much improved.

Is there perhaps an aerial or close up photograph of one of the elephants before necropsy, showing the degree of autolysis (or lack of it) to replace one of the pictures of the spleen in Figure 2?

We have replaced one of the spleen photos in Figure 2 with a photo of one of the elephant carcasses.

We thank Reviewer #2 for their additional constructive comments and hope that the modifications above have addressed all comments satisfactorily.